# Stigma associated with cutaneous and mucocutaneous leishmaniasis: A systematic review

**Hasara Nuwangi[1], Thilini Chanchala Agampodi[1], Helen Philippa Price[2], Thomas Shepherd[3], Kosala Gayan Weerakoon[4‡], Suneth Buddhika Agampodi**  **[5,6‡] ***

**1** Department of Community Medicine, Faculty of Medicine and Allied Sciences, Rajarata University of Sri Lanka, Saliyapura, Sri Lanka, **2** School of Life Sciences, Keele University, Newcastle-under-Lyme, Staffordshire, United Kingdom, **3** School of Medicine, Keele University, Newcastle-under-Lyme, Staffordshire, United Kingdom, **4** Department of Parasitology, Faculty of Medicine and Allied Sciences, Rajarata University of Sri Lanka, Saliyapura, Sri Lanka, **5** Department of Internal Medicine, Section of Infectious Diseases, Yale University School of Medicine, New Haven, Connecticut, United States of America, **6** International Vaccine Institute, Seoul, Republic of Korea

‡ These authors are joint senior authors on this work.
* suneth.agampodi@yale.edu

**Data Availability Statement:** All relevant data are within the manuscript and its Supporting information files.

**Funding:** This research was carried out as part of the program ECLIPSE funded by the National

## Abstract

### Background

Cutaneous (CL) and mucocutaneous leishmaniasis (MCL) are parasitic diseases caused by parasites of the genus *leishmania* leading to stigma caused by disfigurations. This study aimed to systematically review the dimensions, measurement methods, implications, and potential interventions done to reduce the CL- and MCL- associated stigma, synthesising the current evidence according to an accepted stigma framework.

### Methods

This systematic review followed the PRISMA guidelines and was registered in PROSPERO (ID- CRD42021274925). The eligibility criteria included primary articles discussing stigma associated with CL and MCL published in English, Spanish, or Portuguese up to January 2023. An electronic search was conducted in Medline, Embase, Scopus, PubMed, EBSCO, Web of Science, Global Index Medicus, Trip, and Cochrane Library. The mixed methods appraisal tool (MMAT) was used for quality checking. A narrative synthesis was conducted to summarise the findings.

### Results

A total of 16 studies were included. The studies report the cognitive, affective, and behavioural reactions associated with public stigma. Cognitive reactions included misbeliefs about the disease transmission and treatment, and death. Affective reactions encompass emotions like disgust and shame, often triggered by the presence of scars. Behavioural reactions included avoidance, discrimination, rejection, mockery, and disruptions of interpersonal relationships. The review also highlights self-stigma manifestations, including

Institute for Health and Care Research (NIHR - https://www.nihr.ac.uk) (NIHR200135) using UK aid from the UK Government to support global health research (H.N., T.C.A, T.S, H.P.P K.G.W., and S.B.A) The funders had no role in the study design, data collection, and analysis, decision to publish, or preparation of the manuscript. The views expressed in this article are those of the authors and not necessarily those of the NIHR or the UK Department of Health and Social Care.

**Competing interests:** The authors have declared that no competing interests exist.

enacted, internalised, and felt stigma. Enacted stigma manifested as barriers to forming proper interpersonal relationships, avoidance, isolation, and perceiving CL lesions/scars as marks of shame. Felt stigma led to experiences of marginalisation, rejection, mockery, disruptions of interpersonal relationships, the anticipation of discrimination, fear of social stigmatisation, and facing disgust. Internalised stigma affected self-identity and caused psychological distress.

## Conclusions

There are various manifestations of stigma associated with CL and MCL. This review highlights the lack of knowledge on the structural stigma associated with CL, the lack of stigma interventions and the need for a unique stigma tool to measure stigma associated with CL and MCL.

### Author summary

The stigma surrounding cutaneous (CL) and mucocutaneous leishmaniasis (MCL) is multifaceted, encompassing cognitive, affective, and behavioural reactions such as misconceptions, negative emotions like disgust and shame, and discriminatory actions. This stigma is further perpetuated by self-stigmatization, which includes enacted, felt, and internalised stigma. As a consequence, individuals with CL and MCL experience psychological distress, marginalisation, rejection, difficulties in forming interpersonal relationships, and heightened anticipation of encountering discrimination.

Our review reveals several gaps in knowledge, including a lack of understanding regarding structural stigma, insufficient appropriate interventions, and the urgent requirement for a specialised tool to measure the stigma related to CL and MCL. Moreover, the absence of a standardised theoretical framework for stigma research on these conditions has led to inconsistent data generation, emphasising the need for a universal stigma framework applicable to various health conditions. Such a framework would foster a deeper understanding, enabling effective strategic planning to address the impacts of stigma on individuals with CL and MCL.

## Introduction

Leishmaniasis is a neglected tropical disease (NTD) with three main manifestations; cutaneous leishmaniasis (CL), mucocutaneous leishmaniasis (MCL), visceral leishmaniasis (VL), and is endemic in 98 countries across the world [1,2]. An additional form of leishmaniasis, Post Kala-azar dermal leishmaniasis (PKDL) is a skin-related complication of VL, which develops in a subset of patients following recovery from primary disease [3].

Stigma is a complex phenomenon influenced by cultural norms and community beliefs and has various conceptualisations [4]. Erwin Goffman initially introduced the concept as 'the situation of the individual who is disqualified from full social acceptance' [5], and subsequent scholars have defined and conceptualised stigma differently [6–10]. The reasons for the differences are that the concept has been applied to a multitude of unique circumstances, the research on stigma is multidisciplinary, and researchers in different fields have approached the concept from different theoretical dimensions [9]. Weiss identified three major types of stigma

related to NTDs: enacted, anticipated, and internalised [11]. Additionally, scholars have also referred to social stigma, or public stigma, which is defined as 'beliefs held by a sizable fraction of society which places people with the stigmatised condition in a less equal place or a part of an inferior group which creates barriers for affected people' [12]. Concerning leishmaniasis, scholars have used the term 'aesthetic/unesthetic stigma' to describe stigmatisation based on bodily deformities [13–16].

This systematic review is grounded in Bos et al.'s conceptual framework of stigma [17] which encompasses four interrelated stigma manifestations; public stigma, self-stigma, stigma by association, and structural stigma. Public stigma, which is the core component of the framework, refers to people's psychosocial reactions (cognitive, affective, and behavioural) to someone with a perceived stigmatising condition. Cognitive reactions could be misbeliefs or stereotypes. Affective reactions are emotional reactions such as anger or irritation. Behavioural reactions can be rejection avoidance or discrimination [17,18]. Self-stigma, also sometimes referred to as internalised stigma, is the social and psychological impact on individuals with stigmatising characteristics, including the internalisation of negative beliefs and feelings associated with the condition and the fear of stigmatisation [17]. Self-stigma encompasses three types of manifestations impacting from public stigma: enacted stigma (negative treatment of a person with a stigmatised condition), felt stigma (anticipation or experience of stigma), and internalised stigma (psychological distress and reduced self-worth of a person with a stigmatised condition) [17,19,20]. Stigma-by-association is the psycho-social reactions to people associated with a stigmatised person and/or how people react to being associated with a stigmatised person [17,21]. Structural stigma is the legitimisation and perpetuation of a stigmatised status by institutions and ideological systems of society [17,18].

Compared to stigmatising diseases, such as leprosy and tuberculosis, there exists a notable knowledge gap and insufficient synthesis of existing data concerning CL and MCL [22–24]. Stigma can cause health inequalities [25], poor quality of life and mental health issues [10,26–28]. Mental health effects and the psychosocial burden of CL have been systematically reviewed [29,30], while evidence on CL- and MCL- associated stigma remains unclear. Despite being acknowledged as a stigmatising disease [31] there is limited understanding of the specific types of stigma associated with the diseases and their varied implications. It is crucial to understand stigma to develop interventions [32]. This study aimed to systematically review the dimensions, measurement methods, implications, and potential interventions done to reduce the CL- and MCL- associated stigma, synthesising the current evidence according to an accepted stigma framework.

## Methods

This study was developed according to the Reporting Guidelines for Systematic Review and Meta-Analysis (PRISMA) and is registered in the International Platform of Registered Systematic Review and Meta-analysis Protocols PROSPERO (ID- CRD42021274925). The protocol was previously published in PLoS ONE [33].

### Eligibility criteria

**Inclusion criteria.** We included primary articles that discuss any type of CL- and MCL-associated stigma entirely or partially. Only studies published in English, Spanish, and Portuguese were included. Articles up to January 2023 were considered. Qualitative (including ethnographic/anthropological studies), quantitative and mixed-method studies on human leishmaniasis were included.

**Exclusion criteria.** Articles targeting laboratory-based research, clinical trials, diagnostic or treatment methods for CL and MCL, veterinary studies, vector studies, and articles that explore stigma only in VL or PKDL were excluded.

## Search strategy

We performed an electronic search in MEDLINE, Embase, Scopus, PubMed, EBSCO, Web of Science, Global Index Medicus, Trip, and Cochrane Library databases. We manually searched the reference lists of the finally selected articles and identified the articles meeting the inclusion criteria but were initially not detected by our search. Stigma, CL- and MCL-related keywords were used for the search (S1 File).

## Study selection, data collection

Rayyan platform was used to manage references and article inclusion/exclusion [34]. The search results were uploaded to Rayyan, and duplications were removed. Two investigators, including the first author (HN), then independently reviewed the titles and abstracts of all search results and excluded articles in stage one. In stage two, the articles selected for full-text screening were retrieved, and data were extracted, or the paper was excluded. Conflicts were resolved by discussions among authors (HN, TA, KG). The reasons for exclusion were documented (Fig 1).

## Data analysis

Data were extracted and compiled in Excel. Articles in Portuguese and Spanish were translated using DeepL translator [35]. Three authors (HN, TA and KW) discussed and synthesised data in several cycles. A narrative synthesis was done to compile data. The data is presented according to the stigma conceptualisation by Bos et al. [17].

## Study quality and risk of bias

Quality checking was done using MMAT [36]. The authors indicated the appropriateness of the study for particular criteria, as required by the tool. A score was calculated for each article based on recommendations by Pace et al. [37].

# Results

We retrieved 4622 records from nine databases and three from searching reference lists of selected papers. After removing the duplicates, we manually screened 1933 articles. We included sixteen articles for the synthesis in this review (Fig 1). The quality assessment of different studies is provided in S2 File. Out of the 16 articles, five were considered high quality (>80% score), six were of moderate quality (79–60% score) and five were of low quality (<60%). None of the articles were excluded based on study quality (S2 File).

## Characteristics of included papers

The time period of the studies conducted ranged from 1987–2018. The key characteristics of the included articles are detailed in Table 1. The selected studies came from three regions across the world 1) Asia—Afghanistan [38–40], Iran [16], Yemen [15] 2) South America–Brazil [41,42], Colombia [43,44], Suriname [45,46] and 3) Africa—Morocco [47,48], Tunisia [49,50], with four studies conducted in low-income countries (Afghanistan, Yemen), five in lower-middle-income countries (Iran, Tunisia, Morocco), and six in upper middle-income countries (Suriname, Brazil, Colombia) based on the World Bank's country classification by

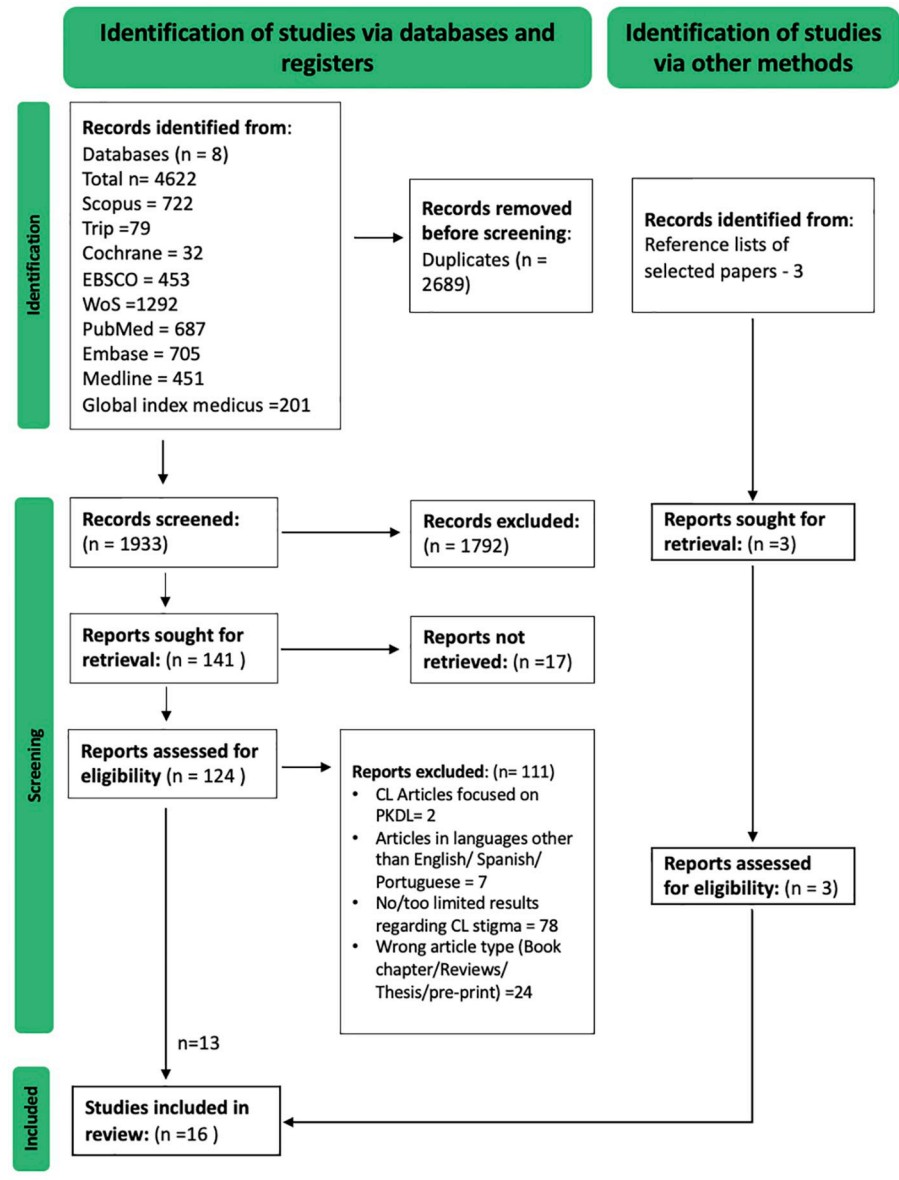

**Fig 1. PRISMA Flowchart.**

income level. One study was a multi-country study based on work carried out in seven countries (Brazil, Burkina Faso, Colombia, Iran, Morocco, Peru, and Tunisia) [51].

## Study designs

Of the sixteen articles, eleven were qualitative [15,16,39–43,45,47,49,51], two were quantitative [44,50], and the rest of the three were mixed methods studies [38,46,48]. Five out of 11 of the qualitative studies have used interviews as a data collection method [16,41,42,49,51]. Three studies used only focus group discussions (FGDs) [38,39,47]. Two studies used both interviews and FGDs [40,43]. Two studies have used a questionnaire with open-ended questions [15,45]. Apart from that, Ramdas et al. have used ethnography as a data collection method [45].

**Table 1. Characteristics of the included papers.**

| | Author/s | English title | CL or MCL | Publication year | Country/s where the study was conducted | Methods | Study design | Data collection methods (and details) | Time period of the research | Sample size/ number of participants |
|---|---|---|---|---|---|---|---|---|---|---|
| 1 | J.M. Costa [42] | Psychosocial and stigmatizing aspects of mucocutaneous leishmaniasis | MCL | 1987 | Brazil | Qualitative | Descriptive cross sectional | Interviews | Not specified | 15 patients with severe MCL and 25 individuals without MCL from interior of Bahia (Três Braços and Corte de Pedra) |
| 2 | Hugh Reyburn [40] | Social and psychological consequences of cutaneous leishmaniasis in Kabul, Afghanistan | CL | 2000 | Afghanistan | Qualitative | Descriptive cross sectional | Interviews and FGDs | Feb -July 1998 | Interviews n = 14, FGDs n = 70, Participants were adult clinic attendees with moderate/ severe CL, usually affecting the face or hands For FGDs, their unaffected spouses also attended from Kabul Afghanistan |
| 3 | Richard Reithinger [38] | Social Impact of Leishmaniasis, Afghanistan | CL | 2005 | Afghanistan | Mixed methods | Descriptive cross sectional | house-to-house survey (HHS) and FGDs | Oct 2002 | Survey– 252 community members from 5 districts of Kabul. 13 FGDs with 108 women from the same communities |
| 4 | Carree C. Stewart and William R. Brieger [39] | Community views on Cutaneous Leishmaniasis in Istalif, Afghanistan: implications for treatment and prevention | CL | 2009 | Afghanistan | Qualitative | Descriptive cross sectional | Focus group discussions | Not specified | 8 FGDs with 6–9 members each from Istalif, Afghanistan in each |
| 5 | Sahienshadebie Ramdas [45] | Nuancing stigma through ethnography: the case of cutaneous leishmaniasis in Suriname | CL | 2016 | Suriname | Qualitative | Ethnography/ Anthropological | Participant observation and short (structured) questionnaire, with open-ended questions | Sept 2009 to Dec 2010 | 205 CL patients, 6 healthcare workers, 321 community members including 18 people with a history of CL from Dermatology Service in Paramaribo, hinterland (Godo-olo, Brokopondo Centrum area, Donderskamp, Tepu) and the Brazilian gold diggers village of Benzdorp |

*(Continued)*

**Table 1.** (Continued)

| | Author/s | English title | CL or MCL | Publication year | Country/s where the study was conducted | Methods | Study design | Data collection methods (and details) | Time period of the research | Sample size/ number of participants |
|---|---|---|---|---|---|---|---|---|---|---|
| 6 | Mohamed Kouni Chahed [50] | Psychological and Psychosocial Consequences of Zoonotic Cutaneous Leishmaniasis among Women in Tunisia: Preliminary Findings from an Exploratory Study | CL | 2016 | Tunisia | Quantitative | Descriptive cross sectional | Interviewer administered questionnaire | Not specified | 41 female CL patients with scars from El Hichria (n = 31) and Ouled Mhamed (n = 10) in the Sidi Bouzid Governorate |
| 7 | Issam Bennis [47] | "The mosquitoes that destroy your face". Social impact of cutaneous leishmaniasis in South-eastern Morocco, A qualitative study | CL | 2017 | South-eastern Morocco | Qualitative | Explanatory case study approach | FGDs | Mar and Apr 2015 | 251 individuals from CL endemic areas in South-eastern Morocco: Errachidia and Tinghir provinces |
| 8 | Issam Bennis [48] | Psychosocial impact of scars due to cutaneous leishmaniasis on high school students in Errachidia province, Morocco | CL | 2017 | Morocco | Mixed methods | Descriptive cross sectional | Self-administered questionnaire | Apr 2015 | 448 students in rural districts of Errachidia province with CL outbreaks between 2008 and 2010 |
| 9 | Sandro Javier Bedoya Pacheco [41] | Social stigmatization of cutaneous leishmaniasis in the state of Rio de Janeiro, Brasil | CL | 2017 | Brazil | Qualitative | Descriptive cross sectional | Interviews based on a semi structured questionnaire | Not specified | 24 CL patients with skin lesions in exposed areas living in leishmaniasis endemic areas in Rio de Janeiro |
| 10 | Mohamed Ahmed Al-Kamel [15] | Stigmata in cutaneous leishmaniasis: Historical and new evidence-based concepts | CL | 2017 | Yemen | Qualitative | Descriptive cross sectional | Questionnaire with an oral component | May 2016 | 11 CL patients from Sana and Radaa |
| 11 | Alireza Khatami [16] | Lived experiences of patients suffering from acute old world cutaneous leishmaniasis: a qualitative content analysis study from Iran | CL | 2018 | Iran | Qualitative | Descriptive cross sectional | Interviews | Oct 2010 to Nov 2011 | 12 CL patients from in CL endemic Kashan region and non-endemic Tehran region |
| 12 | Libardo J. Gómez [44] | Stigma, participation restriction and mental distress in patients affected by leprosy, cutaneous leishmaniasis and Chagas disease: a pilot study in two co-endemic regions of eastern Colombia | CL | 2019 | Colombia | Quantitative | Descriptive cross sectional | Interviews based on four questionnaires | Apr to Jun 2018 | 306 individuals with a diagnosis of leprosy, CL or Chagas disease (CD) from Norte de Santander and Arauca 106 people with leprosy 98 with CL 100 with CD |

(Continued)

**Table 1.** (Continued)

| | Author/s | English title | CL or MCL | Publication year | Country/s where the study was conducted | Methods | Study design | Data collection methods (and details) | Time period of the research | Sample size/ number of participants |
|---|---|---|---|---|---|---|---|---|---|---|
| 13 | Ricardo V. P. F. Hu [46] | Body location of "New World" cutaneous leishmaniasis lesions and its impact on the quality of life of patients in Suriname | CL | 2020 | Suriname | Mixed methods | Descriptive cross sectional | Two quantitative questionnaires and interviews | Jan 2010 to May 2013 | 46 CL patients from administered to Dermatology Service Paramaribo, Suriname |
| 14 | Astrid C. Erber [51] | Patients' preferences of cutaneous leishmaniasis treatment outcomes: Findings from an international qualitative study | CL | 2020 | Brazil, Burkina Faso, Colombia, Iran, Morocco, Peru, and Tunisia | Qualitative | Descriptive cross sectional | Semi-structured in-depth interviews | Not specified | 74 CL patients from endemic regions of Brazil, Burkina Faso, Colombia, Iran, Morocco, Peru and Tunisia |
| 15 | Aicha Boukthir [49] | Psycho-social impacts, experiences and perspectives of patients with Cutaneous Leishmaniasis regarding treatment options and case management/ An exploratory qualitative study in Tunisia | CL | 2020 | Tunisia | Qualitative | Descriptive cross sectional | Semi-structured interviews | Not specified | 10 CL patients from Sidi Bouzid and Gafsa, Tunisia |
| 16 | Robin Van Wijk [43] | Psychosocial burden of neglected tropical diseases in eastern Colombia: an explorative qualitative study in persons affected by leprosy, cutaneous leishmaniasis and Chagas disease | CL | 2021 | Colombia | Qualitative | Descriptive cross sectional | FGDs and semi structured interviews | May to Jun 2018 | All participants were adults (≥18 years old) with a diagnosis of leprosy, CL or CD from Norte de Santander or Arauca, FGD- 4 FGDs with a total of 34 individuals, Interviews– 13 individuals |

CL–Cutaneous leishmaniasis; MCL–Mucocutaneous leishmaniasis; FGD- Focus group discussion

## Study participants

Of the 16 studies, 14 included people with CL/MCL [15,16,40–51], and 7 studies included community members without the disease [38–40,42,45,47,48] (Table 1). Of the 16 articles, 15 were solely focused on CL [15,16,38–41,43–51], and one on MCL [42].

The study done by Chahed et al. was done with female CL patients only [50], and the study done by Reithinger et al. had a study component focussing on FGDs done with women in the selected communities [38]. All the other studies had both male and female participants.

### Synthesis of evidence on the stigma associated with CL and MCL

**Stigma types recorded in the studies.** Different authors have explored different stigma types associated with CL and MCL. Here we present a comprehensive account of the stigma types authors have reported in their respective studies (Table 2).

The social stigma was the commonest manifestation reported. The social stigma associated with CL was present in 8 studies conducted in 6 countries; Morocco [47,48], Brazil [41], Tunisia [49,50], Colombia [44], Yemen [15] and Afghanistan [40]. Enacted stigma was reported in Iran [16], Tunisia [50], and Afghanistan [40]. Self/internalised stigma was reported in Iran [16] and Morocco [48]. Perceived/felt stigma was reported in Iran [16] and Afghanistan [40]. Aesthetic stigma was reported in Iran [16] and Yemen [15]. Al-Kamel introduces three types of CL-related stigma; CL Social stigma, CL aesthetic stigma and CL psychological stigma [15].

Four studies of this systematic review mention that stigma is present but have not specified which type of stigma prevails with MCL [42] or CL [38,39,52]. Table 2 shows a comprehensive account of the studies and different stigma manifestations.

**Stigma tools.** Six tools were used to measure stigma and/or to make topic guides for interviews and FGDs (Table 2). The two quantitative studies have used the revised Illness Perception Questionnaire (IPQ-R), Psoriasis Life Stress Inventory (PSLI) questionnaire and World Health Organization Quality of Life-26 (WHOQOL-26) [50] and Explanatory Model Interview Catalogue (EMIC) [44]. Hu et al. have used the stigma assessment guidelines by the International Federation of Anti-leprosy Associations (ILEP) and the Netherlands Leprosy Relief (NLR) [46] to draft their study tool. Out of the six tools, the EMIC is the only instrument that directly measures stigma.

**Stigma manifestations.** Stigma manifestations were synthesised and categorised into public stigma, self-stigma, and stigma by association (Fig 2).

**Public stigma.** The synthesised evidence on cognitive, affective, and behavioural reactions of public stigma is reported in Table 3. Various misbeliefs about the disease were identified as cognitive reactions of public stigma. Disgust and shame were the main negative emotional reactions associated with CL- and MCL- associated public stigma. Avoidance, discrimination, rejection, mockery and disruption of interpersonal relationships are the types of behavioural reactions reported in the studies.

**Self-stigma.** Public stigma affects the self in three main ways; enacted, internalised, and felt (Fig 2). People with CL and MCL encountered all three main types of self-stigma. Manifestations of each stigma type are described below.

*Enacted stigma.* We identified three main manifestations of enacted stigma. Each of the manifestations is described below.

Barriers to forming proper interpersonal relationships. Studies done in Morocco [47,48] show that CL scars on the face are a barrier to marriage. This has affected women more than men. Participants have stated they will not let their sons marry a woman with CL scars. However, CL scars were not a reason for divorce. Studies by Stewart & Brieger and Reithinger et al. in Afghanistan [38,39] show that women with a lesion or a scar will face the threat of not finding a husband because of CL. However, in Tunisia, in a study done by Chahed et al. [50] with women, most participants have reported that CL scar reduces the marriage prospects of men (75%) more than women (59%). The disease has affected the interpersonal relationships of women with CL scars in all spheres of life; family, social and professional. Reithinger et al. show that in Afghanistan [38], the social role of women as a mother/wife may be severely affected by this disease. In an FGD, 21 out of 96 participants (22%) have said that a mother with CL should be prevented from breast-feeding her child; 48 (51%) have stated that they would prevent a person with CL from touching or hugging

**Table 2. Details about stigma associated with CL and stigma tools used as reported in the articles.**

| | Authors | Country/s where the study was conducted | Does the study gives evidence about presence of stigma? | If Yes what types of stigma? | What is the tool used to measure/ discover stigma? | Information reported regarding the validity and reliability of the tools used in the study |
|---|---|---|---|---|---|---|
| 1 | Costa et al., 1987 [42] | Brazil | Yes | Have not categorized stigma into different stigma types | Interviews guide—Guide comprised of three parts, which questioned aspects of the patient's life, before, during and after treatment | Whether interview guide was piloted or changed during the study was not mentioned |
| 2 | Bennis, Belaid, et al., 2017 [47] | South-eastern Morocco | Yes | Social stigma | A topic guide inspired by Brown et al. [56] | After completion of the first two FGD, they have adapted the topic guide added two new questions: 'How can you make the scars go away?' and 'How do people in general behave with those affected with this disease?' |
| 3 | Hu et al., 2020 [46] | Suriname | Study mentions that there's no stigma | N/A | For HRQL assessment -Skindex-29 questionnaire and EQ-5D-3L questionnaire For illness experience assessment–semi-structured questionnaire | Illness experience semi structured questionnaire—The exploratory inquiries were partly grafted on the stigma assessment guidelines, developed by the International Federation of Anti-leprosy Associations (ILEP) and the Netherlands Leprosy Relief (NLR) |
| 4 | Stewart & Brieger, 2009 [39] | Afghanistan | yes | Have not categorized stigma into different stigma types. | Interview guide | In-depth interviews were conducted with key informants of the community for design of the focus group discussion guide. |
| 5 | Khatami et al., 2018 [16] | Iran | Yes | Enacted stigma Perceived (felt) stigma, Internalized (self) stigma, Aesthetic or unaesthetic stigma | Interview guide | Not specified |
| 6 | Ramdas et al., 2016 [45] | Suriname | Study mentions that there's no stigma | N/A | Questionnaire with open ended questions | Not specified |
| 7 | Gómez et al., 2020 [44] | Colombia | Yes | Social stigma | Explanatory Model Interview Catalogue (EMIC) for stigma | Internal consistency of the questionnaire has been tested by, Cronbach's Alpha ($\alpha = 0.85$). |
| 8 | Erber et al., 2020 [51] | Brazil, Burkina Faso, Colombia, Iran, Morocco, Peru, and Tunisia | Yes | Social stigma | Interview guide | An adapted translated version of interview topic guide developed in collaboration of all investigators participating in the larger study was used. |
| 9 | Boukthir et al., 2020 [49] | Tunisia | Yes | Social stigma | Interview guide | An adapted translated version of interview topic guide developed in collaboration of all investigators participating in the larger study was used. |
| 10 | Chahed et al., 2016 [50] | Tunisia | Yes | Social stigma and anticipated stigma | Evaluating illness perception: Revised Illness Perception Questionnaire (IPQ-R) Assessing psychosocial adjustment to stress from skin disease: Psoriasis Life Stress Inventory (PSLI) questionnaire Assessing quality of life: World Health Organization Quality Of Life-26 (WHOQOL-26) scale | "Revised Illness Perception Questionnaire (IPQ-R)—The questionnaire has been translated and adapted to ZCL and cultural characteristics of Sidi Bouzid Psoriasis Life Stress Inventory (PSLI) questionnaire—The questionnaire, is translated, back-translated and adapted, for use on patients with ZCL |

(*Continued*)

**Table 2.** (Continued)

| | Authors | Country/s where the study was conducted | Does the study gives evidence about presence of stigma? | If Yes what types of stigma? | What is the tool used to measure/ discover stigma? | Information reported regarding the validity and reliability of the tools used in the study |
|---|---|---|---|---|---|---|
| 11 | van Wijk et al., 2021 [43] | Colombia | Yes | The authors declared that there are no indications of stigma associated with CL. However, we identified self-stigma experiences reported in the results | FGD guides Interview guide | The interviews consisted of five predefined questions based on Weiss's framework for the assessment of health-related stigma [11] |
| 12 | Bennis, Thys, et al., 2017 [48] | Morocco | Yes | Self-stigma, Social stigma | self-administered questionnaire with quantitative and qualitative questions | Questionnaire was pre-tested with 10 students from Errachidia city |
| 13 | Reyburn et al., 2000 [40] | Afghanistan | Yes | Enacted stigma, Felt stigma Social stigma | FGD guides | Not specified |
| 14 | Reithinger et al., 2005 [38] | Afghanistan | Yes | Have not categorized stigma into different stigma types. | Survey questionnaire—House-to-house survey (HHS), FGDs—The same house-to-house survey (HHS) | The authors report the questionnaire survey used was standardized. |
| 15 | Pacheco et al., 2017 [41] | Brazil | Yes | Social stigma | Semi structured questionnaire | Not specified |
| 16 | Al-Kamel, 2017 [15] | Yemen | Yes | The author has identified specific CL related stigma and is termed as below, CL Social stigma CL aesthetic stigma CL psychological stigma | Survey questionnaire | Not specified |

their children; 55 of 96 participants (57%) said that a person with CL must not cook for the family.

Avoidance, isolation and marginalisation. Studies show evidence of others avoiding close contact with CL patients and isolating or marginalising them. According to Reithinger et al., in their study done in Kabul, Afghanistan [38], 40 out of 89 (46%) FGD participants have stated that they will isolate a person with CL, keeping themselves away from the person and even using separate items from the person with CL. The study done by Stewart & Brieger in Istalif, Afghanistan [39], also shows that others will avoid and isolate a person with CL. The same study reported that children with CL were prevented from attending school and playing, men with CL could not work, and women with CL could not carry out household work. According to Khatami et al., in Iran, people with moderate or severe CL are rejected and isolated in public [16].

CL lesion/scar is seen as a mark of shame. In Morocco, people see the CL scar as a mark of shame. They explain that people with CL do not have the same appearance as before and are different from others [48]. In Afghanistan, CL is seen as something that brings shame to the whole family [40].

*Felt stigma*. Felt stigma is the experience or anticipation of stigmatisation by a person with a stigmatised condition [17]. There were eight manifestations of felt stigma, as described below.

Experiencing marginalisation and rejection. The study done by Costa et al. in Bahia, Brazil [42], reports that 11 out of 15 (73%) people with MCL felt a feeling of marginalisation, and 9 out of 15 (60%) people reported that others moved away from them. Some patients with severe MCL have even self-isolated themselves.

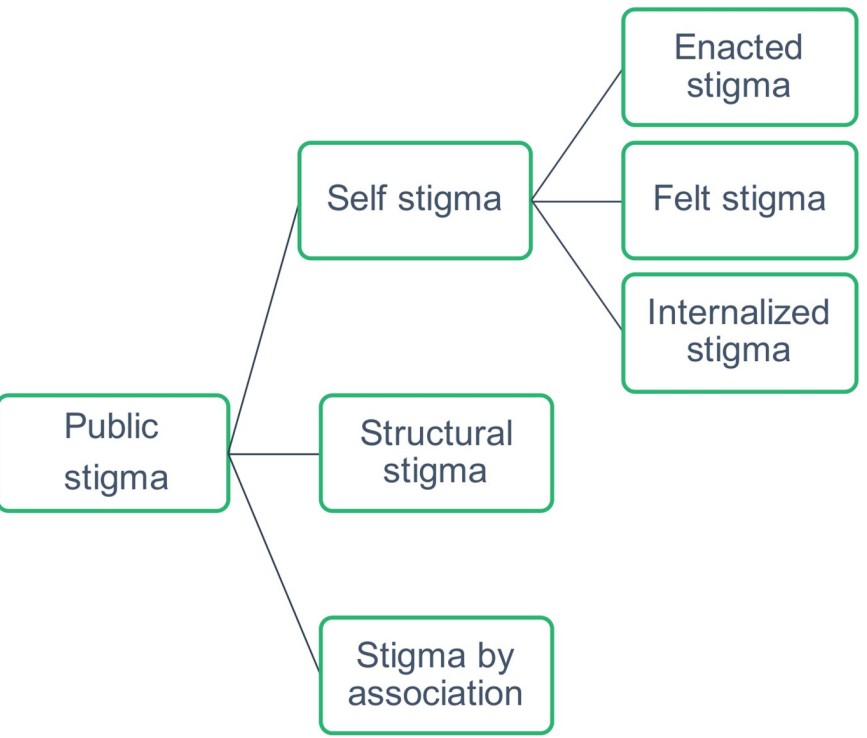

**Fig 2. The stigma conceptualisation used in this study.**

Facing rejection was a common issue that people with CL faced. According to studies done in Tunisia [49,50], women with CL scars have described their feeling of isolation. They also experience rejection and consider the scar as a source of rejection. In Tunisia, the study done by Chahed et al. [50] shows that rejection experiences and anticipation and avoidance of stress were significantly and negatively correlated with age.

Across the selected studies, people with CL faced rejection in the following ways: avoiding close contact, not looking at a CL patient [16], the tendency to stare at the scar [48], and exclusion [40]. The scar was a potential source of rejection [49], and the psychological status of people with CL was affected by rejection [48]. In Kabul, Afghanistan, men face more rejection in public as within the context of the country's culture, men interact more with society [40]. In Suriname [46], people with CL tend to keep a distance from others in order to avoid uncomfortable situations as they anticipate negative remarks from others. However, authors attribute this to facing negative reactions, not necessarily stigma. In Colombia [44], people living in rural areas have experienced higher participation restrictions (p = 0.037). In Morocco [48], negative behaviours from others lead to people with CL feeling isolated, which results in them struggling with everyday life.

<u>Facing mockery and ridicule.</u> The study by Costa et al. in Bahia, Brazil [42], shows that MCL patients have experienced being ridiculed by others. In Errachidia province, Morocco [48], people with CL fear ridicule by others. In Afghanistan [40], names like "spotty", and "Saldanee" are used to address people with CL [39], and CL patients are faced with mockery by their own family members.

<u>Affecting the occupation.</u> Studies done in Tunisia by Boukthir et al. [49] and Chahed et al. [50] reported that a visible scar is a barrier to finding employment. In Afghanistan, people

**Table 3. Cognitive, affective and behavioural reactions of public stigma.**

| | Cognitive reactions | |
|---|---|---|
| **Types of reactions** | **Different reactions documented in studies** | **References** |
| **Misbeliefs about disease transmission** | • In Errachidia province, Morocco, the belief that CL could be transmitted person-to-person was a major cause of social isolation and rejection<br>• And in southern Morocco, people were reluctant to share a meal with a person with an open CL lesion because of fear of infection<br>• In Iran, there were instances where even healthcare workers believed that this is a highly transmissible disease (person-to-person) and refused to treat people with CL<br>• In Tunisia, people have misconceptions about the mode of transmission of the disease, and people believe that it is highly contagious<br>• In Afghanistan, public stigma manifested because of the belief that CL could be spread by talking to a person with CL<br>• In Afghanistan, there was a misbelief among the participants that the disease could be transmitted by physical touch. Out of 360 respondents, 86 mentioned "touching", and 26 mentioned "sharing meals and household goods" as a mode of transmission<br>• In Yemen reports misbeliefs about were causes of stigma | Bennis, Thys, et al., 2017 [48]<br>Bennis, Belaid, et al., 2017 [47]<br>Khatami et al., 2018 [16]<br>Boukthir et al., 2020 [49]<br>Reyburn et al., 2000 [40]<br>Reithinger et al., 2005 [38]<br>Al-Kamel, 2017 [15] |
| **Misbeliefs about the treatment side effects** | The study done by Erber et al. Showed that people fear that treatment could cause infertility in men | Erber et al., 2020 [51] |
| **Misbeliefs about mortality of the disease** | In Yemen the belief that the disease can lead to death, was a causes of stigma | Al-Kamel, 2017 [15] |
| | Affective reactions | |
| **Disgust** | • In Afghanistan, disgust is a reason for the stigmatisation and exclusion of people with CL<br>• According to Boukthir et al., disgust arising from scars is a potential source of rejection of CL patients in Tunisia | Reyburn et al., 2000 [40]<br>Boukthir et al., 2020 [49] |
| **Shame** | • Two studies done in Morocco show a degree of shame associated with living in an area with CL and others see the scar as a mark of shame<br>• People in Afghanistan believed that a person with CL brings shame to the family | Bennis, Belaid, et al., 2017 [47] and Bennis, Thys, et al., 2017 [48]<br>Reyburn et al., 2000 [40]. |
| | Behavioural reactions | |
| **Avoidance** | • In Brazil people with MCL faced avoidance and noted a distance between them and others<br>• In Suriname, some people anticipated negative remarks, distanced themselves from others, and experienced avoidance. However, in that paper, the authors state that there is no stigma attached to CL in Suriname and that most people have not experienced negative reactions from others | Costa et al., 1987 [42].<br>Hu et al., 2020 [46] |
| **Discrimination** | • Costa et al. Report that in their study done in Brazil, 73.3% of participants have faced discrimination<br>• In Iran and Morocco CL patients have faced discrimination.<br>• In Kabul, Afghanistan women with CL have even experienced domestic violence | Costa et al., 1987 [42]<br>Khatami et al. [16], 2018; Bennis, Belaid, et al., 2017; Bennis, Thys, et al., 2017 [47,48]<br>Reyburn et al., 2000 [40] |

with CL have reported facing jeering in the streets, which has led them to fear going to work [40], and they believe that the CL lesion harms the personality and that men cannot work properly because of that [39].

Others questioning about the disease. The study by Khatami et al. in Iran [16] shows that questioning about the disease by others and having to explain the origin of the disease have made people feel discouraged and upset. People have resorted to lying about the disease in order to avoid questions. Erber et al. [51] also discuss that others questioning about the disease has caused people to feel embarrassed and stigmatised.

Experiencing disruption of interpersonal relationships. CL was a cause of disruptions in interpersonal relationships, including marriage [15,39,40,47–49,51]. Both men and women with MCL [42] and CL [51] have faced difficulty in forming relationships with the opposite

sex. In Afghanistan, Khatami et al. report instances of women with facial lesions facing violence from their husbands [40]. In Brazil, Costa et al. [42] report that three out of 15 people with severe MCL lesions and their families were marked by the communities. These people with MCL have faced difficulties forming relationships with the opposite sex. According to Al-Kamel, girls with CL lesions in Yemen will not be able to marry [15].

Anticipating discrimination. Both MCL and CL patients anticipate discrimination by others [16,42,47,48]. A study participant in Morocco has said that "CL patients will be unable to cope with society due to fear of social discrimination and contempt" [48]. However, another study in Morocco mentions that discriminating attitudes decreased over time [47]. According to Khatami et al. [16], CL patients in Iran were aware of others who have faced discrimination and were afraid of facing the same consequences.

Fear of social stigmatization. An international study done in Brazil, Burkina Faso, Colombia, Iran, Morocco, Peru, and Tunisia [51], and a study by Bennis, Thys et al. in Morocco [48] shows that people with CL expressed fear of social stigmatisation.

Facing Disgust. In Afghanistan, people with CL were disgusted by others. The authors mention disgust as a main factor for stigmatisation [40].

*Internalised stigma*. Internalised stigma is the reduction of self-worth and experiencing psychological distress due to a reduction in self-worth by a person with a stigmatised condition [17]. Internalised stigma manifestations are described below.

Effect on the self-identity. Stigma has an effect on the self-identity of people with CL. van Wijk et al. report a case of a person with CL in Eastern Colombia [43], who did not feel equal to others because of the level of disability accompanying the disease and in Errachida province, Morocco [48], people perceive that the disease affects the masculinity of men. In Tunisia, women with CL scars believe that they have an "impaired identity" and do not feel like "fully fledged individuals" [50].

Devaluation due to body image concerns. People with MCL in Bahia, Brazil [42] felt ashamed of their bodies. Studies show that people with CL considered CL as a deformity [16,38,39], which resulted in anger [39] and feeling discomfort about their appearance [41]. Studies done in Morocco [48] and Tunisia [49,50] show that people with CL scars were considered less attractive. In Iran [16], people were self-disgusted about their own lesions. This has led to them feeling sad. In Tunisia [49], people have said that their scars are ugly and a potential source of disgust. In Morocco [48], people believe that affected people will disgust themselves and can even end up hating themselves.

Diminished self-esteem. A study conducted with women in Tunisia found a relationship between emotional representations and the loss of self-esteem [50]. In Morocco, people with CL fear meeting others, leading to self-contempt [48].

*Stigma by association*. Several studies found that people have experienced stigma, rejection [15,44,47] and shame [47] related to the geographical area with CL. In Morocco, people have noted that they are ashamed to live in an endemic area for CL and often feel rejected by relatives living abroad who have contracted CL during visits to Morocco [47].

In Brazil, Costa et al. [42] report an incident where the family members of a young girl with MCL faced struggles in finding life partners.

*Structural stigma*. None of the studies explored the potential structural stigma associated with CL and MCL.

*Coping mechanisms*. Some studies reported what mechanisms people have used to cope with the stigma and rejection they face from others. The coping mechanisms are listed in Table 4.

Fig 3 is an illustration of the systematic review findings regarding stigma, aligning with the stigma concept presented by Bos et al.

**Table 4. Various coping mechanisms adopted by CL and MCL patient.**

| Category | Coping mechanisms |
|---|---|
| **1 Emotion-focused coping** | |
| 1.1 Taking on the blame | In Iran people tend to take on the blame and admitted their own role in getting the disease and feeling that it is their responsibility [16] |
| 1.2 Tolerating/understanding negative reactions from others | People with CL in Iran tend to either tolerate or understand the reactions towards them. They rationalize this by "trying to see the disease through the eyes of others" and accepting their situation [16]. In Afghanistan people have resorted to accepting the isolation they face [40] |
| 1.3 Spiritual factors | In Iran people with CL resorted to praying to God, asking for a quick recovery [16]. In Morocco, people believe that one gets because of "God's will" and "destiny". They believe that God decides who is contracting the disease and who is healing [48]. In Afghanistan, CL is seen as a punishment from God and a person gets CL because of their sins [40] In Yemen people have dealt with shame believing that the disease was given by God [15] |
| **2 Problem-focused coping** | |
| 2.1 Hiding the lesion/scar or self | In Morocco women who fear social stigmatization resort to hiding the lesion [47]. In Iran people use strategies such as wearing long sleeved clothes. They have resorted to hiding affected body parts and isolating themselves [16]. In Afghanistan, women with CL were pleased to have the burqa as it gives them a chance to hide, however, it is difficult to use this as a coping strategy within the home with the family [40] |
| 2.2 Modifying social interactions | People with CL in Iran used modified social interactions as a coping strategy [16]. In Tunisia people with CL developed anticipatory avoidance behaviour against the social rejection they face [50] |
| 2.3 Using makeup | An international qualitative study done in seven countries by Eber et al. show that women with CL have used make up as a coping mechanism [51]. In Morocco girls have used skin creams to conceal the lesion [48] |

*Vulnerable populations.* The stigma associated with CL and MCL was more pronounced in certain populations than in others. Table 5 categorises the available evidence according to the affected population. Women, young people, people with severe/multiple or facial lesions and people from rural areas were identified as vulnerable populations.

*Mental health implications of stigma.* The diminished mental and emotional well-being of a person. In Iran, people with CL have faced complex psychological issues, such as anger and distress from mistreatment, worry about scarring, and sadness from disgust towards lesions. Anger stemming from the disease experience and how patients are treated has affected the interpersonal relationships of people with CL [16].

The study by Bennis, Thys et al. in Morocco [48] also shows that people with CL experience adverse psychological suffering and suicidal ideations. In Kabul, Afghanistan, people with the disease feel disappointed, sad, and angry due to lack of kindness and family rejection. The disease leaves them disempowered and vulnerable to silencing. They have experienced emotional isolation. Participants with children with CL (n = 83) have stated that CL cause trauma (n = 45, 54%) because of the disfiguration caused by lesions or scars (n = 20), and because of exclusion from playing with other children (n = 6) [38]. In Yemen, a study reports that psychological stigma associated with CL is the most prevalent CL stigma type [15].

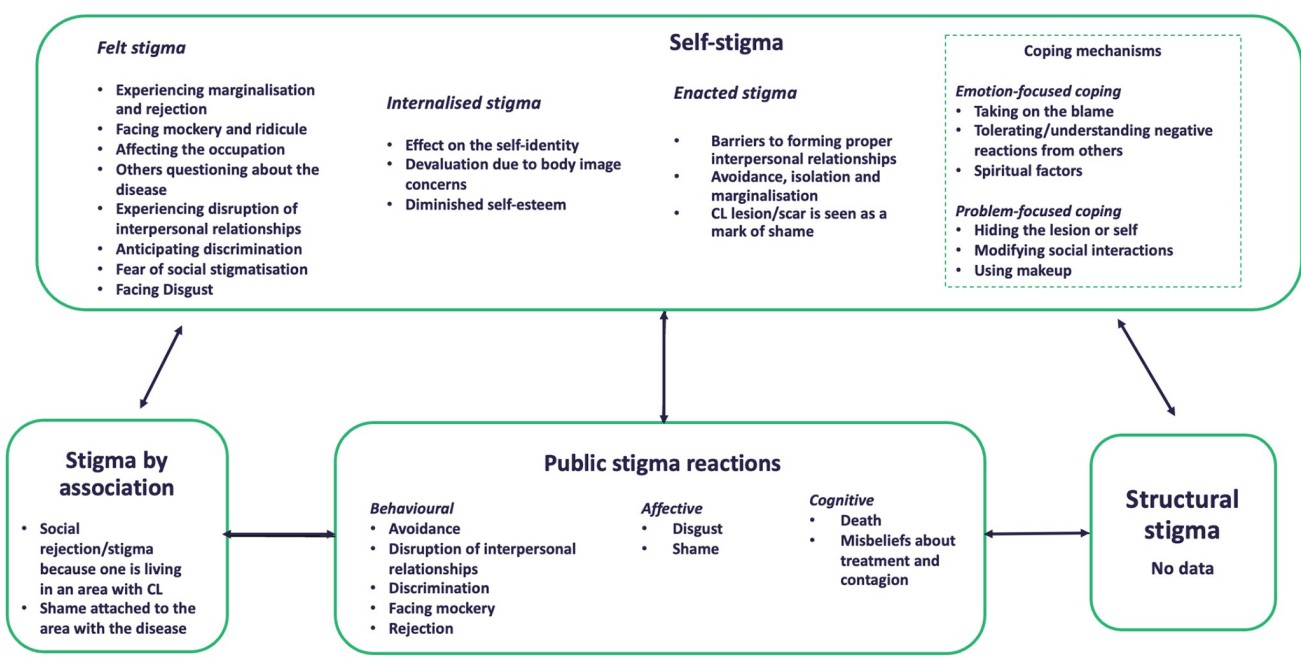

**Fig 3. Diverse manifestations of CL- and MCL-associated stigma.**

Being ashamed is another negative feeling that was reported. Reports from Iran [51], and Yemen [15] suggested that people felt ashamed of their facial lesions, and women in Tunisia expressed shame due to CL scars [50]. Similar sentiments of shame and embarrassment were also reported in studies conducted in Morocco [48], and Afghanistan [40].

**Table 5. Details about the vulnerable populations across studies.**

| Population affected | Country/s of origin and references | Details reported in the studies |
|---|---|---|
| **Women** | Morocco [47,48] Afghanistan [39,40] Iran [16] Tunisia [49,50] Yemen [15] | Studies show CL scar affected women more than men [51] especially with regard to marriage. |
| **Young people** | Tunisia [50] | In Tunisia, stigma is more pronounced in some age groups, and young people have experienced more stigma |
| **People with severe/ multiple/facial lesions** | Morocco [47], Tunisia [49,50] Afghanistan [38] | People with severe forms of CL or with multiple/facial lesions are more affected by stigma |
| **People from rural areas** | Colombia [44]. Yemen [15] | In Norte de Santander and Arauca, Colombia, EMIC scores significantly differed between rural and urban areas (p<0.001). The median EMIC score for rural areas was 6, and for urban areas, 0. In rural areas, people affected by CL anticipated/perceived a higher level of stigma and experienced significantly higher levels of participation restriction. In a study by Al-Kamel in Yemen, all participants who experienced stigmatisation were from rural areas. One female participant with a small CL lesion on the nose from an urban setting has reported that she has not experienced any stigma as she believes the disease was given by God. |

*Absence of stigma.* Two studies done in Suriname concluded that there was no stigma related to CL [45,46]. In Suriname, the Health-related quality of life (HRQL) impact was similar between those with facial lesions and lesions on other body parts, and the size or number of lesions did not correlate with enacted stigma.[46]. Surinamese individuals know CL is not contagious, and there is no evidence of discrimination or enacted stigma. Ramdas et al. report that aesthetic stigma was absent, particularly among male participants who were less concerned about scars. The absence of (enacted) stigma was attributed to the nature of CL in Suriname, which does not cause severe facial disfigurations. However, some individuals did face negative reactions that ceased after the lesions healed, possibly indicating self, internalised, or anticipated stigma. These aspects were not thoroughly examined in the study [45].

A study in Colombia evidence shows that the disease is considered common and normal, and people do not try to conceal the disease. Some people have faced negative attitudes, such as avoidance due to fear of contagion, only for a certain period [43]. In Yemen, where stigma was sometimes very significant, old age was a reason for the absence of social and aesthetic stigma [15].

## Discussion

The main finding of our review was that there is public and self-stigma (felt, internalised and enacted) associated with CL and MCL in different countries. This review also categorises the stigma associated with CL and MCL based on existing theories and frameworks, providing a foundation for future research.

This review shows the numerous implications and manifestations of the stigma associated with CL and MCL which exists globally. In countries like Afghanistan [38–40], the stigma associated with CL is more prominent, while in Suriname, stigma is almost non-existent [45,46]. There is a lack of evidence about the stigma in South Asian countries where the disease is endemic [53–55].

The included studies discuss the stigma associated with both scars [38–40,47–50] and lesions [16,38,39,49]. In stigmatising skin diseases, severity and location or visibility are key drivers of stigma [56]. Hence, Further research is needed to understand the different implications of stigma based on the presence of scars, active lesions, or after the disease is cured. People have experienced a range of stigma manifestations pertaining to felt stigma, enacted, and internalised stigma. Felt stigma manifested as experiencing marginalisation and rejection [16,40,47,48], facing mockery and ridicule [39,42,48], affecting the occupation [39,40,49,50], others questioning about the disease [16,51], experiencing disruption of interpersonal relationships [15,39,40,47–49,51]. They have anticipated discrimination by others [16,42,47,48], and fear of social stigmatisation [48,51]. Enacted stigma experiences were barriers to forming proper interpersonal relationships [16,38,39,47,48], avoidance, isolation and marginalisation [16,38,39] and CL lesion/scar being considered a mark of shame [40,48]. Effect on the self-identity [43,48,50], devaluation due to body image concerns [16,38,39,41,42,48–50] and diminished self-esteem [48,50] were internalised stigma experiences. These stigma implications are similar to those seen in other stigmatised skin diseases like leprosy [57] and psoriasis [58]. Health-related stigma is considered a hindrance to the prevention and control of diseases [59,60]. Stigma could either deter health-seeking and obstruct treatment adherence or, in some rare cases, improve adherence and enhance treatment-seeking behaviour [11]. The impact of CL- and MCL-associated stigma on the health-seeking process is poorly understood. There is a need to study the consequences of CL and MCL-associated stigma on health-seeking behaviour to develop effective public health interventions.

Women, young people, and people from rural areas are at a higher risk of experiencing more profound stigma. Similar findings have been observed in diseases like leprosy [61]. However, in Afghanistan, men have faced unique suffering in public because of the socio-cultural context [40]. Studies done in Colombia show that rural people perceived/anticipated a higher level of stigma and experienced more participation restrictions [44]. Further exploration is needed to understand if rural people experience higher levels of stigma within their community or from external sources. It is also important to investigate how stigma impacts vulnerable populations and the underlying reasons for potential variations.

The review identifies several unexplored areas related to stigma in CL and MCL. Although CL is considered a disease among the poor [62], the studies on the stigma associated with CL and MCL do not reflect the influence of structural determinants on the disease. None of the reviewed articles mentions "structural stigma" or the social and institutional ideologies contributing to CL- and MCL- MCL-associated stigma. Structural stigma implications of other diseases, such as leprosy, are well documented [60].

These studies do not explore whether there are established stereotypes about CL or MCL that lead to stigma. Stereotypes are a major component of the cognition of stigma [9,18]. Whether there are established stereotypes attached to CL and MCL should be explored to better understand CL- and MCL- associated stigma.

Stigma interventions should aim to address and interrupt the stigma process before it is applied [32]. To do that, drivers, facilitators and manifestations of stigma should be researched and understood properly. One of the significant gaps we must highlight is the lack of interventions on stigma. The lack of interventions for CL and MCL stigma could be due to inadequate understanding and a knowledge gap. We also wish to highlight the need for new studies as some of the included studies are a few decades old and could be outdated [42].

Another major finding is the lack of quantitative studies done on CL- and MCL- associated stigma and specific tools to measure stigma for these conditions. The currently used tools are inadequate to measure the unique stigma associated with CL and MCL [44,46,50]. EMIC is a 4-point Likert scale measuring stigma [63]. Higher scores indicate more stigma. However, in the study conducted by Gómez et al., the validated tool is not provided as supplementary information, making it challenging to assess the tool's accuracy in measuring stigma related to CL [44]. Chaded et al. have used PSLI, IPQ-R and the WHOQOL as tools in their study [50]. PSLI is a self-rating tool with 15 items evaluating psoriasis-related stress [64]. The stigma section of PSLI focuses on potential stressful events related to cosmetic disfigurement and social stigma, such as feeling self-conscious in public. IPQ-R is a four-level Likert scale exploring various domains, including illness identity, timeline, consequences, cure, control, coherence, and emotional representation [65]. WHOQOL scale assesses an individual's holistic health, incorporating physical and mental well-being [66]. PSLI, IPQ-R, and WHOQOL do not fully measure or quantify stigma, nevertheless, they provide insights into its consequences. There is an urgent need to develop a tool unique to CL and/or MCL that can be adapted to different cultural contexts. For Leprosy [67] and HIV [68] such tools are available.

The main strength of this review lies in its thorough examination of studies related to the stigma associated with CL and MCL. By encompassing diverse study types and their outcomes within a well-established stigma framework, the results presented here can be effectively utilized by various stakeholders. This study has several limitations. Non-English/Spanish/Portuguese articles were excluded, and only online sources were considered, potentially leading to the omission of grey literature. Further, the data and conceptualization presented here remain relevant to the selected papers and the defined scope outlined in the paper.

The concept of stigma in the context of CL and MCL studies lacks consistent grounding in a selected theoretical framework, resulting in inconsistent data [9]. To address this, future

research should employ a theoretical framework and clearly define the concepts used. More-over, we draw attention to the complex nature of stigma as a concept and stress the necessity for a more comprehensive stigma framework that can be applied to comprehend diverse manifestations of stigma across various health conditions. This would enhance understanding of the diverse manifestations of stigma and enable the development of effective strategies to address its impact. Furthermore, it is essential to accurately identify the specific manifestation of stigma when reporting a study rather than using the term "stigma" arbitrarily. By doing so, a more nuanced understanding can be achieved, leading to targeted interventions tailored to the unique contextual expressions of stigma [45,69].

## Conclusion

In conclusion, this systematic review demonstrates the presence of diverse stigma manifestations linked to CL and MCL in certain contexts while highlighting the absence of stigma in other contexts suggesting that stigma associated with CL and MCL is not universal. The findings also emphasise the link between stigma and misunderstandings regarding disease transmission and its consequences. The findings highlight the need for further research on structural stigma, stigma interventions and a dedicated stigma assessment tool for these conditions.

## Supporting information

**S1 File. Search results.**
(PDF)

**S2 File. Risk of bias assessment.**
(XLSX)

## Acknowledgments

The authors thank Dr Brianne Wenning, Mr Ahmedh Aaqil Rifky, Mr Sandaru Hasaranga Shanthapriya and the ECLIPSE team for supporting the work presented here.

## Author Contributions

**Conceptualization:** Hasara Nuwangi, Thilini Chanchala Agampodi, Helen Philippa Price, Thomas Shepherd, Kosala Gayan Weerakoon, Suneth Buddhika Agampodi.

**Data curation:** Hasara Nuwangi.

**Formal analysis:** Hasara Nuwangi.

**Funding acquisition:** Helen Philippa Price.

**Methodology:** Hasara Nuwangi, Thilini Chanchala Agampodi, Helen Philippa Price, Thomas Shepherd, Kosala Gayan Weerakoon, Suneth Buddhika Agampodi.

**Project administration:** Suneth Buddhika Agampodi.

**Supervision:** Thilini Chanchala Agampodi, Helen Philippa Price, Thomas Shepherd, Kosala Gayan Weerakoon, Suneth Buddhika Agampodi.

**Visualization:** Hasara Nuwangi.

**Writing – original draft:** Hasara Nuwangi.

**Writing – review & editing:** Hasara Nuwangi, Thilini Chanchala Agampodi, Helen Philippa Price, Thomas Shepherd, Kosala Gayan Weerakoon, Suneth Buddhika Agampodi.

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
