## [Decision Letter · Decision Letter 0]

24 Oct 2023

Dear Prof Agampodi,

Thank you very much for submitting your manuscript "The Stigma Associated with Cutaneous and Mucocutaneous Leishmaniasis: A Systematic Review" for consideration at PLOS Neglected Tropical Diseases. As with all papers reviewed by the journal, your manuscript was reviewed by members of the editorial board and by several independent reviewers. The reviewers appreciated the attention to an important topic. Based on the reviews, we are likely to accept this manuscript for publication, providing that you modify the manuscript according to the review recommendations. 

Sincerely,

Felix Bongomin, MB ChB, MSc, MMed, FECMM

Academic Editor

Charles Jaffe

Section Editor

Reviewer's Responses to Questions

**Key Review Criteria Required for Acceptance?**

**Methods**

-Are the objectives of the study clearly articulated with a clear testable hypothesis stated?

-Is the study design appropriate to address the stated objectives?

-Is the population clearly described and appropriate for the hypothesis being tested?

-Is the sample size sufficient to ensure adequate power to address the hypothesis being tested?

-Were correct statistical analysis used to support conclusions?

-Are there concerns about ethical or regulatory requirements being met?

Reviewer #1: The methods are clear and articulate; appropriate design, sufficient sample size and correct statistical analyses were employed. The study was registered.

Reviewer #2: The objectives of the study arre clearly articulated and the study design is appropriate to address the stated objectives

**Results**

-Does the analysis presented match the analysis plan?

-Are the results clearly and completely presented?

-Are the figures (Tables, Images) of sufficient quality for clarity?

Reviewer #1: The analysis plan matches the actual analysis; results clearly presented; however the figures attached in the appendices seem blury,it would be good to use clearer figures as in the body of the manuscrpit

Reviewer #2: The results are clearly and completely presented. Figures and tables included are sufficient

**Conclusions**

-Are the conclusions supported by the data presented?

-Are the limitations of analysis clearly described?

-Do the authors discuss how these data can be helpful to advance our understanding of the topic under study?

-Is public health relevance addressed?

Reviewer #1: The conclusion is subtle and clearly describes a summary of the data, limitations, public health implications of the study.

Reviewer #2: The conclusions are supported by the data presented

**Editorial and Data Presentation Modifications?**

Reviewer #1: No recommendations to submit

Reviewer #2: Minor revision: A discrepancy is found between the articles included for the synthesis referred to in the text as mentioned in the comments to the authors.

**Summary and General Comments**

Reviewer #1: -I am concerned that this same study has already been published as below https://doi.org/10.1371/journal.pone.0285663

http://www.ncbi.nlm.nih.gov/pmc/articles/pmc10174477/

-Could there be any reason why this paper is undergoing a review for publication again? or could this be an issue of ethical concern

Reviewer #2: Comments to the authors

The bibliographic search article includes an appropriate methodology and all articles included in the manuscript are referenced. Nevertheless, a discrepancy is found between the included articles for the synthesis referred to in the text (16 articles, lines 157, 158, 177, …) and those of studies included in the review in Figure 1 (20 articles).

Specific comments

Abstract

Do not include abbreviations in the summary, or do so for all concepts introduced (MMAT, ...).

Introduction

Include a more recent WHO reference regarding the clinical manifestations of leishmaniasis and its geographical distribution. In any case, as it is in general, it should not be limited, like the one included (ref. 1), to a cutaneous leishmaniasis reference nor to the eastern Mediterranean region. Also, usually three main clinical manifestations of leishmaniasis are considered: cutaneous, mucocutaneous and visceral. PKDL is a known complication of VL, as LCD or other complications are from LC. Perhaps because of this, the authors do not include studies that specifically deal with PKDL.

Lines 99 and 103: In line 99 it is mentioned the study of Bos et al. (ref. 15), in relation to the stigma types related to the impact of public stigma, but at the end of the phrase (line 103) there are included three references (15,16,17). Should it be replaced “Bos et al.” by “Some authors”?

Methods – Eligibility criteria

Change “Qualitative, quantitative, cross-sectional, mixed-method, and ethnographic/anthropological studies were included, but articles targeting veterinary studies, vector studies, laboratory-based research, clinical trials, diagnostic or treatment methods for CL and MCL and articles that explore stigma only in VL or PKDL were excluded.” by “Qualitative, quantitative, cross-sectional, mixed-method, and ethnographic/anthropological studies on human leishmaniasis were included, but articles targeting laboratory-based research, clinical trials, diagnostic or treatment methods for CL and MCL and articles that explore stigma only in VL or PKDL were excluded.”

Include some explanation on the types of studies: qualitative, quantitative, cross-sectional, mixed-method, and ethnographic/anthropologica.

Study design

- Line 178: is reference 37 included among the mixed methods studies?

- Line 179: Replace “The majority” by “Five out of 11”

Results

- Table 1 and 2: include the reference number next to the author, as done in the table 3.

- Figure 3: The font size of the figure could be enlarged?

- Starting from line 245 of the results, and to improve tracking, include in the text references next to the authors and remove the references from the end of the sentence. Follow the examples, line 245: Studies by Bennis, Thys, et al. in Morocco [45,46] … for divorce.” and line 248 “Studies by Stewart & Brieger [37] and Reithinger et al. [36] in Afghanistan …. because of CL”.

- Line 408: Replace “Multiple” by “Two”.

Discussion

Given that the authors previously published an article on a protocol for a systematic review of the stigma associated with cutaneous leishmaniasis and mucocutaneous leishmaniasis, it would be of interest to include some discussion of the gaps and difficulties encountered in the application of said protocol in the bibliographic search just carried out and how they could be solved.

References

- Include complete reference for reference 16.

- Review the reference 22.

- Reference 38 cannot be consulted with the information included.

PLOS authors have the option to publish the peer review history of their article (what does this mean?). If published, this will include your full peer review and any attached files.

Reviewer #1: No

Reviewer #2: No

Figure Files:

Data Requirements:

Reproducibility:

References

---

## [Editor Report · Decision Letter 1]

16 Nov 2023

Dear Prof Agampodi,

Thank you very much for submitting your manuscript "The Stigma Associated with Cutaneous and Mucocutaneous Leishmaniasis: A Systematic Review" for consideration at PLOS Neglected Tropical Diseases. As with all papers reviewed by the journal, your manuscript was reviewed by members of the editorial board and by several independent reviewers. The reviewers appreciated the attention to an important topic. Based on the reviews, we are likely to accept this manuscript for publication, providing that you modify the manuscript according to the review recommendations. 

There are additional editor comments to be addressed.

Sincerely,

Felix Bongomin, MB ChB, MSc, MMed, FECMM

Academic Editor

Charles Jaffe

Section Editor

There are additional editor comments to be addressed.

Figure Files:

Data Requirements:

Reproducibility:

References

---

## [Editor Report · Decision Letter 2]

24 Nov 2023

Dear Prof Agampodi,

We are pleased to inform you that your manuscript 'Stigma Associated with Cutaneous and Mucocutaneous Leishmaniasis: A Systematic Review' has been provisionally accepted for publication in PLOS Neglected Tropical Diseases.

Best regards,

Felix Bongomin, MB ChB, MSc, MMed, FECMM

Academic Editor

Charles Jaffe

Section Editor

---

## [Editor Report · Acceptance letter]

18 Dec 2023

Dear Prof Agampodi,

We are delighted to inform you that your manuscript, "Stigma Associated with Cutaneous and Mucocutaneous Leishmaniasis: A Systematic Review," has been formally accepted for publication in PLOS Neglected Tropical Diseases.

Best regards,

Shaden Kamhawi

co-Editor-in-Chief

Paul Brindley

co-Editor-in-Chief
